# Impact of enterprise digital transformation on capacity utilization: Evidence from China

**Ning Zhao**, **Jianxin Ren***

School of Business Administration, Zhongnan University of Economics and Law, Wuhan, China

* renjianxin@zuel.edu.cn

**Data Availability Statement:** All relevant data are within the manuscript and its Supporting Information files.

**Funding:** This study was supported by the National Natural Science Foundation of China (Grant No.71903201).

## Abstract

Based on empirical analysis of Chinese listed companies from 2010 to 2018, we demonstrate that enterprise digital transformation has a significant impact on improving capacity utilization. Digital transformation is a significant driving force behind enterprise-specific production and innovation. Furthermore, enterprise innovation and enterprise-specialized production play a mediating role in the impact of enterprise digital transformation on capacity utilization. Based on these baseline findings, heterogenous analysis reveals that the impact of digital transformation on capacity utilization is significant for firms with larger capital scales or poor governance and manufacturing abilities. However, it is less important for enterprises with small- and medium-sized capital scales or with more standardized governance, as well as non-manufacturing (service) enterprises.

## Introduction

Digitalization is a major driving force for the future development of the global economy. In comparison to traditional economies, digital economies transcend the limitations of physical economic development factors and fundamentally alter the operational logic and value-adding methods of business organizations [1]. As an important focal point for the development of digital economies, enterprise digital transformations adopt cloud computing, big data, artificial intelligence, and other digital technologies to empower various industries and fields, which have become core drivers for promoting quality and efficiency. Based on the importance of digital transformation, the economic effects of digitalization are gradually receiving increasing attention.

Some studies have focused on the impact of information and communication technology on capacity utilization, arguing that IT applications make firm production more accurate and flexible, reduce information asymmetry between supply and demand, promote technological innovation, reduce external transaction costs, enhance information sharing, and improve production flexibility, which in turn increases resource utilization efficiency [2, 3]. However, a lack of technology, obsolete equipment, unstable power supply, and weak regulations can reduce the effectiveness of information and communication technology [4], as well as capacity utilization.

**Competing interests:** NO authors have competing interests The authors have declared that no competing interests exist.

Several early studies explained the factors influencing capacity utilization in terms of market competition. Some researchers [5–8] have argued that the "excessive entry theorem" explains the problem of duplication and overcapacity in oligopolistic markets. This theorem claims that when the market is open to new entrants and Cournot competition is enforced, the number of firms in equilibrium will exceed the number that maximizes social welfare. Another stream of literature indicates that excess capacity is an important competition strategy that pre-entrants use to create credible threats and entrance barriers [8–11]. Previous studies have explored the causes of overcapacity from the perspective of the reconfiguration of firm operating models and business forms, as well as the combinability of digital technologies, which has resulted in the reconfiguration and nested linking of traditional factors of production [12] and the formation of mutually empowering endogenous systems between digital technologies and traditional factors. Such systems have introduced profound changes into organizational management and production models [13] and have improved the accuracy of corporate investment and production decisions, thereby reshaping the corporate value creation model [14] and necessitating an examination of changes in corporate capacity utilization from a digital perspective. According to this theoretical perspective, although Chinese local governments actively promote digital transformations, we suspect that most enterprises struggle with low capacity utilization rates and overcapacity. Therefore, we used data from Chinese listed companies to investigate whether digital transformation will improve capacity utilization empirically.

This study makes three important contributions to the literature. First, unlike most previous studies that discuss capacity utilization theoretically at the macro and local government levels, this study examined capacity utilization empirically at the micro firm level. Second, in terms of our research design, we analyze the capacity utilization of enterprises by treating the levels of specialized production and firm heterogeneity as important mediating variables, and a comparative analysis is then performed to demonstrate the heterogeneous impacts of digital transformation on capacity utilization. This is a novel approach because these two factors have never been used to explain capacity utilization in previous studies. Third, while most previous studies on the role of digital economies have aimed at reducing resource mismatch and increasing productivity, we aimed to support the proposal that the progress of digital transformation should be combined with real enterprise development, which may broaden the research dimensions of digital economies. Although this paper uses the text analysis method of machine learning to measure the overall situation of enterprise digital transformation, it has not yet realized the accurate depiction of the digitalization of various links within the enterprise (such as research and development, production, sales, etc.).

The remainder is organized as follows. The following section provides a theoretical analysis of the impact of digital transformation on enterprise capacity and presents our research hypotheses. The next section then presents an econometric model and describes the construction of indicators and data sources. Empirical analysis, robustness tests, mechanism tests, and heterogeneity analysis are presented in the following two sections. The final section concludes this paper with a broader discussion of our findings and their policy implications.

## Theory and hypotheses

The capacity utilization rate, which is also known as the equipment utilization rate, is the ratio of total industrial output to production equipment, which represents how much of the theoretical production capacity is actually in use. When an enterprise's output is significantly lower than its normal capacity or its product supply significantly exceeds the market demand, enterprise capacity is poorly utilized. Therefore, effective output optimization and a reduction in the degree of resource mismatch are critical for alleviating overcapacity.

Enterprise digital transformation improves technological innovation capabilities, optimizes production and operational processes, and reduces internal and external communication costs, thereby providing a feasible path to alleviating corporate overcapacity [3]. In terms of innovation models, digital technology enables timely interaction between internal and external innovation agents, allowing cutting-edge technologies to capture large amounts of consumer behavior data to guide corporate innovation [15]. Enterprises can also improve information exchange and collection with inaccessible external organizations through the internet, allowing them to obtain more resources for adopting effective innovation strategies [16, 17]. Firms with sufficient big data processing and analysis capabilities can improve their organizational learning, which can help them innovate and develop new products [18]. According to Porter and Heppelmann [19], the arrival of the digital technology revolution has triggered many innovations that improve product performance and functionality, thereby enhancing organization operational performance. Consequently, efficient production increases competitive advantage, breaks the product homogeneity dilemma, generates new consumer demand, captures a larger market share, and increases firm capacity utilization.

Yuan et al. [20] revealed that digitalization promotes firm specialization by lowering external transaction costs and that the effect of digitalization on firm specialization is stronger for firms with low internal control costs, as well as firms in manufacturing and competitive industries. Furthermore, they found that digitalization increases total factor productivity by expanding firm boundaries, which has a positive impact on economic efficiency. Additionally, labor specialization increases productivity, drives technological innovation, and promotes firm economic growth [21], which improves resource utilization and reduces the level of overcapacity. This is because digital transformation exerts a greater impact on reducing external transaction costs than it does on reducing internal control costs. In other words, the role of driving the movement of corporate boundaries toward a specialized division of labor is dominant. Furthermore, firm production shifts resources from low-utilization sectors to high-utilization sectors, and the horizontal expansion of firms boost capacity utilization. Therefore, we propose the following hypothesis:

**H1**: Enterprise digital transformation increases the level of specialization in production and innovation to increase capacity utilization. In other words, specialization and product innovation act as mediators in the path of influence.

Although previous studies have examined the impact of digital transformation on enterprises, only a few have focused on capacity utilization, particularly the negative impact of digital transformation on business development. Although digital transformation through the advancement of digital technology and infrastructure has already begun, there are still considerable uncertainties regarding how it will be integrated into existing corporate organizations [22]. As a type of change, digital transformation is not always linear in its progression. Excessive emphasis on digital transformation while overlooking its integration with existing production models and strategic options may be counterproductive to improving capacity utilization for the following reasons. First, because the integration of digital technology and industrial systems is still in its early stages, such "exploratory innovation" is highly risky. Therefore, overzealous digital transformation in the early stages will not only consume existing production resources but also create additional resource mismatches. Crozet et al. [23] proposed regulatory indicators for digital services and found that the use of enhanced digital services has a significant negative impact on data-intensive manufacturing. In the early stages of digital transformation, the lack of a suitable institutional design to promote the process will fail to improve the actual efficiency of resource utilization, and transformation will essentially lead to optimizing the IT base, rather than overall synergy, which impedes capacity utilization

improvement. According to Peng [24], the negative impact of digital technology on energy consumption lies in the indirect effects on economic growth caused by the energy plundering effect. For example, in the communications industry, digital transformation is accompanied by huge energy consumption for the production, use, and disposal processes of related products. Lange et al. [25] contended that digitalization has both inhibitory and enhancing effects on energy consumption. The inhibitory effect is manifested through digitalization's ability to reduce the intensity of resource inputs by improving resource utilization efficiency and optimizing industrial structure. The enhancing effect is the direct increase in energy consumption caused by the production, use, and disposal processes of digital technologies, as well as the indirect energy demand caused by the development of a digitally enabled economy. In summary, excessive energy consumption has a negative impact on capacity utilization.

Currently, digital transformation processes are hampered by laborious data governance and "data silos." In the strategic planning of certain enterprises, digital strategy and business development have "two lines and two skins," whereas digital transformation and business development have a weak correlation. Therefore, we propose the following hypothesis:

**H2:** The effect of digital transformation on capacity utilization exhibits an inverted U shape.

## Sample selection and data

The rapid expansion of China's digital economy corresponds to the rapid development of digital technology. The trend of widespread use of digital tools began after 2009. Therefore, in this study, we considered A-share non-financial listed companies from 2010 to 2018 as an initial research sample. Relevant financial and governance data were primarily obtained from the Wind and CSMAR databases, which were mainly sourced from the China City Statistical Yearbook in previous years. Company innovation and innovation-input-related data were compared to Wind database data to check for gaps, and data from different databases were verified by reviewing company annual reports when inconsistencies appeared in the calibration process.

To ensure the quality of our findings, the sample was screened in the following ways: (1) excluding companies with special treatment statuses, (2) excluding companies with abnormal financial statuses, (3) excluding samples with missing relevant data, and (4) tailing at the 1% level for relevant important variables by year. Following the screening process, the sample consisted of 15,942 annual company observations. During the data collection process, Python was used for text mining and Stata16 was used for data processing.

## Key variables

**Explained variable.**   *Capacity utilization rate (CU)*. This is the ratio of actual output to output capacity, which is commonly used as a measurement index to study the level of overcapacity. Among existing capacity measurement methods, the peak, cost function, production function, and data envelopment analysis methods have been adopted by various scholars. The production function method is a metric based on the definition of technical capacity. Technical capacity is the level of capacity at which all capital stock, labor, land, and other factors of production invested in industrial production can be fully utilized for maximum output, assuming that preferences and technologies are identified. The annual series of capacity data is estimated from the actual application of the production function method for capacity measurement by defining a production function and extrapolating the annual series of actual outputs to its "boundary" using a measurement method. The production function method was used in this study to calculate the capacity utilization rate of each industry and its three main

product segments. This method was adopted for three main reasons. First, the production function method is based on neoclassical growth theory, which is not based on historical experience and has a strong theoretical foundation, good validity, and reliability for measuring capacity utilization. Second, unlike the peak method, the production function method incorporates the influences of technological progress, capital, and labor on outputs, as well as the degree of contribution of each production factor to outputs, which can reflect the changes in and characteristics of economic structures and production activities. Third, the production function method requires only data related to the three indicators of capital, labor, and output for practical application and has certain advantages over the cost function method in terms of data availability. Given that different production links in each industry's industrial chain involve different subsectors, the maximum production capacity that each production link can achieve varies. Therefore, the production function approach can eliminate production capacity differences between different enterprises in each production chain, making measurements comparable and estimation results directly applicable to existing capacity assessment standards.

The Cobb-Douglas production function was used in this study to model the production function in its basic form as follows:

$$Y_{i,t} = f\left(K_{i,t}, L_{i,t}\right) = A_i K_{i,t}^{\alpha} L_{i,t}^{\beta} e^{-\mu}, i = 1, 2, 3, t = 1, 2, \ldots, T,$$

where $i$ denotes different links in the chain, $t$ is the number of sample years, Y is the value of real output expressed as annual business revenue, $K_{i,t}$ is the fixed capital stock expressed as annual average net fixed assets, $L_{i,t}$ is the labor input represented by the annual number of employees, and $A$ denotes the technology level, which is typically a fixed constant. The parameters $\alpha$ and $\beta$ denote the output elasticity values of the fixed capital stock and labor input, respectively. Assuming constant returns to scale, we have

$$0 < \alpha, \beta < 1 \,\&\, \alpha + \beta = 1.$$

The logarithm of the above equation is

$$lnY_{i,t} = \alpha lnK_{i,t} + \beta lnL_{i,t} + lnA - \mu.$$

By extrapolating the production function to its boundary, the boundary production function is obtained as

$$lnY_{i,t}^{*} = \alpha lnK_{i,t} + \beta lnL_{i,t} + lnA,$$

where $Y_{i,t}{}^{*}$ denotes the theoretical maximum level of output or potential output. Let $lnA = \alpha$ and $E(\mu) = \varepsilon$. Substituting these variables into the above equation yields

$$lnY_{i,t} = \alpha lnK_{i,t} + \beta lnL_{i,t} + (\alpha - \varepsilon) - (\mu - \varepsilon).$$

Because $E(\mu - \varepsilon) = 0$, estimating the above equation using ordinary least squares yields

$$ln\hat{Y}_{i,t} = \hat{\alpha} lnK_{i,t} + \hat{\beta} lnL_{i,t} + (\alpha - \hat{\varepsilon}).$$

The boundary production function generated by changing the constant term in the above equation is called the average production function and is further treated as

$$Max(lnY_{i,t} - ln\hat{Y}_{i,t}) = Max\left\{ lnY_{i,t} - \left[ \hat{\alpha} lnK_{i,t} + \hat{\beta} lnL_{i,t} + (\alpha - \hat{\varepsilon}) \right] \right\}.$$

Therefore, the boundary production function derived from the estimation is

$$\hat{Y}_{i,t} = e^{\hat{z}} K_{i,t}^{\hat{\alpha}} L_{i,t}^{\hat{\beta}},$$

Capacity utilization is measured as

$$CU = Y_{i,t}/\hat{Y}_{i,t}.$$

The capacity utilization rate estimated in this manner has a value of one. In other words, the capacity utilization rate obtained for each listed company is not an absolute value. The maximum capacity utilization rate of the selected sample during the period from 2010 to 2018 is 100%, and the remaining estimates are relative to the maximum value during the selected period. Because the maximum capacity that each production link in the industry chain can achieve varies, this treatment eliminates company differences and makes estimation results directly applicable to existing capacity evaluation criteria.

## Explanatory variable

*Degree of digitalization (DT).* Given that a standard metric for measuring the level of firm digital transformation has yet to be developed, we used Python to mine digital transformation data from the annual reports of listed companies. According to a review of existing studies on digital transformation, enterprise digital transformation is measured by capturing the use of information technology such as the internet by enterprises through text analysis [26], but such indicators are unidimensional and focus only on the application of a specific information technology, making it difficult to obtain a comprehensive picture of enterprise digitalization. In contrast, we developed a relatively complete digital lexicon using semantic representations of national digital economy policies and employed machine-learning-based text analysis methods to construct a more comprehensive indicator that reflects the degree of digitalization of Chinese listed enterprises. This digital lexicon was then expanded to a Chinese lexicon to conduct text analysis on the "Management Discussion and Analysis" (MD&A) sections of the annual reports of the listed companies based on machine learning to obtain the frequency of digitalization-related terms in these sections. Given that the length of the MD&A sections of annual reports varies between companies, after extracting the frequency of each keyword in each listed company's annual report, the sum of the frequency of digitalization-related words divided by the length of the MD&A sections of annual reports was used to measure the DT of micro enterprises. This indicator was then multiplied by 100 for notational convenience. The higher the value of the DT indicator, the greater the degree of digitalization.

## Mediating variables

*Firm innovation (Inn).* The data for corporate innovation in this study were primarily derived from the number of invention patent applications in the CSMAR database's patent sub-database of listed companies and subsidiaries. We adopted logarithmic processing to ensure data consistency. Data on corporate innovation and innovation investments were compared to Wind database data to identify gaps, and the data were verified by consulting the annual reports of companies when inconsistencies appeared during the calibration process. Corporate innovation was measured based on the Chinese patent database for the following reasons. First, it ensures better data integrity because the CSMAR database matches with the Wind database to form long-term panel data. Second, it ensures greater data reliability because patent data is based on corporate innovation behavior, as opposed to "R&D investment" or "new product output value." Finally, it facilitates more detailed classification because it assists in

distinguishing between different types of enterprise innovation and refining enterprise innovation behavior based on information from different patent applications and licenses. It is noteworthy that patent data contains additional information such as application, authorization, and patent citation information. Given that this study focused on the impact of digital transformation on innovation behavior, the fluctuation in the number of patent applications should reflect this focus. Therefore, the number of patent applications for inventions was used to measure innovation behavior.

*Degree of production specialization (VSI)*. Vertical integration and specialization are diametrically opposed concepts in labor studies, with higher vertical integration implying lower specialization and vice versa. The value added to sales (VAS) measure was proposed by Adelman [27] as a proxy measure of vertical integration and has been widely used in management studies. The basic concept is to estimate the VAS revenue ratio, where a higher ratio indicates greater firm vertical integration. Following previous studies [28, 29], we adopted a modified value-added method to estimate the VAS as follows:

$$VAS = \frac{\text{Value added} - \text{net profit after tax} + \text{normal profit}}{\text{Income from main business} - \text{net profit after tax} + \text{normal profit}} =$$

$$\frac{\text{Value Added} - \text{Net Income After Tax} + \text{Net Assets} \times \text{Average Net Asset Margin}}{\text{Income from main business} - \text{net income after tax} + \text{net assets} \times \text{Average net asset margin}}.$$

Value added is expressed as the difference between a firm's sales and purchases, and VAS measures value added as a percentage of sales revenue, with larger values indicating greater vertical integration. Following Zhang [30], the inverse of VAS is defined as the VSI. Therefore, VSI is calculated as

$$VSI = 1 - VAS.$$

A higher VSI indicates a higher level of enterprise specialization. Following Fan Ziying and Fei [29], observations in which the VSI deviates from the reasonable value domain (0, 1) interval were excluded from our analysis to ensure this metric's validity.

## Other variables

Many factors contribute to firm overcapacity. In addition to the digital transformation of firms, characteristics such as firm size (*size*), gearing (*lev*), return on total assets (*roa*), and geographical and industry characteristics such as the nature of firms (*soe*) can have an impact on a firm's capacity utilization. Therefore, we controlled for these variables. Furthermore, *code*, *industry*, *province*, and *year* denote firm-, industry-, region-, and year-fixed effects, respectively. Table 1 summarizes the definitions of all the key variables used in this study.

## Empirical model

Following previous studies [20, 29, 31], the impact of digital transformation (inverted "U" mechanism), production specialization, enterprise innovation, and capacity utilization were evaluated through theoretical analysis. To test our hypotheses, the quadratic terms of the explanatory variables were incorporated and the following models were developed:

$$CU_{i,t} = \alpha_0 + \alpha_1 DT_{i,t} + \gamma Controls + \varepsilon_{i,t}, \tag{1}$$

$$CU_{i,t} = \alpha_0 + \alpha_1 DT_{i,t} + \alpha_2 DT_{i,t}^2 + \gamma Controls + \varepsilon_{i,t}. \tag{2}$$

**Table 1. Variable definitions.**

| Variable type | Name | Symbols | Measurements |
|---|---|---|---|
| Explained variable | Capacity utilization rate | CU | Actual and potential outputs are calculated using the production method (revenue from main business over maximum expected revenue) |
| Explanatory variable | Digitalization of enterprises | DT | Text analysis of the annual reports of listed companies using Python to estimate the degree of digital transformation |
| Mediating variables | Corporate innovation | Inn | Logarithm of the number of invention patent applications by listed companies in the year |
| | Production specialization | VSI | The modified value-added method is used to obtain the firm's value added as a proportion of its sales revenue, measuring the firm's VAS, the inverse (1 − VAS) of which indicates the level of specialization |
| Control variables | Company scale | size | ln (Number of employees) |
| | Gearing ratio | lev | (Total liabilities at the end of period / total assets) × 100% |
| | Total return on assets | roa | (Total profit−(income tax rate × total profit)) / (average total liabilities + average owner's equity) × 100% |
| | Nature of business property | soe | If the company is a state-owned enterprise, the value is one. Otherwise, it is 0 |
| | Age of the company | age | ln (Year of observation + 1 −Year of registration) |
| Fixed effects | Region | province | Province of the company |
| | Industry | industry | Industry categories |
| | Year | year | Current year |
| | Company | code | Stock code of listed companies |

If the coefficients $\alpha_1$ and $\alpha_2$ are significantly positive, it indicates that an enterprise's digital transformation promotes its capacity utilization and that its capacity increases in tandem with digital transformation. However, if $\alpha_1$ is significantly positive and $\alpha_2$ is significantly negative, it indicates that the impact of the enterprise's digital transformation on its capacity utilization has an inverted U shape, thereby supporting H2. If $\alpha_2$ is insignificant, it indicates that the impact of enterprise digital transformation on capacity utilization is linear.

Following Wen and Ye [32], the effects of digital transformation, production specialization, firm innovation, and capacity utilization were tested using mediating variables. H1 is tested using the following models:

$$VSI_{i,t} = \alpha_0 + \alpha_1 DT_{i,t} + \gamma Controls + \sum Firm + \sum Year + \varepsilon_{i,t}, \tag{3}$$

$$CU_{i,t} = \alpha_0 + \alpha_1 DT_{i,t} + \beta VSI_{i,t} + \gamma Controls + \sum Firm + \sum Year + \varepsilon_{i,t}, \tag{4}$$

$$Inn_{i,t} = \alpha_0 + \alpha_1 DT_{i,t} + \gamma Controls + \sum Firm + \sum Year + \varepsilon_{i,t}, \tag{5}$$

$$CU_{i,t} = \alpha_0 + \alpha_1 DT_{i,t} + \beta Inn_{i,t} + \gamma Controls + \sum Firm + \sum Year + \varepsilon_{i,t}, \tag{6}$$

$$CU_{i,t} = \alpha_0 + \alpha_1 DT_{i,t} + \theta VSI_{i,t} + \beta Inn_{i,t} + \gamma Controls + \sum Firm + \sum Year + \varepsilon_{i,t}. \tag{7}$$

## Results

### Descriptive analysis

As shown in Table 2, the mean and standard deviation of CU are 1.855 and 5.157, respectively, with a maximum value of 219.8 and minimum value of 0.02, indicating that the degree of capacity utilization varies significantly between enterprises. DT has a mean and standard deviation of 0.229 and 0.353, respectively, with median, maximum, and minimum values of 0.082, 3.637, and 0, respectively. This also indicates that the degree of

**Table 2. Descriptive statistics results.**

| Variable | N | mean | p50 | Sd | min. | max. | range |
|---|---|---|---|---|---|---|---|
| CU | 15,942 | 1.855 | 0.960 | 5.157 | 0.0190 | 219.8 | 219.8 |
| DT | 15,942 | 0.229 | 0.0820 | 0.353 | 0 | 3.637 | 3.637 |
| DT$^2$ | 15,942 | 0.177 | 0.00700 | 0.570 | 0 | 13.23 | 13.23 |
| Inn | 10,766 | 2.260 | 2.197 | 1.463 | 0 | 9.108 | 9.108 |
| VSI | 15,942 | 0.549 | 0.561 | 0.212 | 0 | 1 | 1 |
| roa | 15,942 | 0.0400 | 0.0380 | 0.0750 | −1.648 | 4.837 | 6.485 |
| size | 15,942 | 7.710 | 7.616 | 1.324 | 2.565 | 13.22 | 10.66 |
| lev | 15,942 | 0.427 | 0.420 | 0.211 | 0.00700 | 0.995 | 0.988 |
| age | 15,942 | 2.057 | 2.197 | 0.894 | 0 | 3.367 | 3.367 |
| soe | 15,942 | 0.398 | 0 | 0.489 | 0 | 1 | 1 |

digitalization varies significantly between enterprises. The mean and variance of VSI are 0.549 and 0.212, respectively, with maximum and minimum values of one and zero, respectively, indicating significant differences in the level of specialization between enterprises. Furthermore, the statistical results by industry indicate that there are significant differences between industries in terms of digitalization development with the following characteristics: (1) rapid development of digitalization in the service sector; (2) better integration of manufacturing and digital technology is particularly noticeable in technology-intensive industries; and (3) agriculture, public utilities, and resource-intensive traditional industries exhibit significantly lower digital transformation than the general industry average. Furthermore, there are significant differences in capacity utilization across industries with more capital-intensive industries having lower capacity utilization and technology-intensive industries having higher capacity utilization. This is consistent with the conventional view of overcapacity.

## Baseline regression results

Table 3 presents baseline regression results, where the first column represents controlling only for fixed effects and the second column includes firm- and region-level control variables. The results indicate that the DT coefficients are all significantly positive at less than 1%, implying that the greater the degree of digitalization, the higher the capacity utilization. In terms of economic significance, the results in the second column indicate that a 1% increase in a firm's degree of digital transformation increases its capacity utilization by 31.5%. This suggests that, both economically and statistically, digital transformation has a significant enhancing effect on enterprise capacity utilization. H1 is supported by these empirical results. As stated in the previous theoretical discussion, firm capacity utilization is determined by the difference between actual output and potential maximum output, and the digital transformation of firms may either increase the potential output, thereby exacerbating overcapacity, or increase the actual output level, thereby enhancing firm capacity utilization. The results in Table 3 indicate that digital transformation has a significantly greater effect on actual output than potential output and that enterprise digital transformation significantly increases capacity utilization level, which supports H1. To verify H2, a quadratic treatment of the explanatory variables was performed to obtain the variable DT$^2$. The findings in the third column of Table 3 reveal that the DT$^2$ coefficient is significantly negative. When combined with the coefficient of the primary term, it is evident that the impact of digital transformation on capacity utilization is nonlinear with an inverted U shape.

**Table 3. Benchmark regression results.**

| Variables | (1) | (2) | (3) |
|---|---|---|---|
|  | CU | CU | CU |
| DT | 0.315** | 0.437** | 0.93*** |
|  | (0.137) | (0.18) | (0.338) |
| DT$^2$ |  |  | −0.283* |
|  |  |  | (0.164) |
| size2 |  | −1.053*** | −1.054*** |
|  |  | (0.065) | (0.065) |
| roa |  | 1.936*** | 1.923*** |
|  |  | (0.432) | (0.432) |
| lev |  | 2.693*** | 2.703*** |
|  |  | (0.311) | (0.311) |
| age |  | −0.34*** | −0.345*** |
|  |  | (0.106) | (0.106) |
| soe |  | -0.141 | -0.138 |
|  |  | (0.24) | (0.24) |
| Constants | 1.727*** | 9.406*** | 9.359*** |
|  | (0.087) | (0.516) | (0.517) |
| Observations | 15942 | 15912 | 15912 |
| Adj-R$^2$ | 0.673 | 0.673 | 0.673 |

**Note**: t-statistics adjusted for firm-level clustering are in parentheses; ***, **, and * indicate significance at the 1%, 5%, and 10% levels, respectively (the same is true for the following tables if not otherwise specified).

## Robustness tests

**Endogenous problems.** Previous studies have raised endogeneity concerns for the following two reasons: (a) increased digital transformation of enterprises will improve their production and business environment, and promote higher levels of capacity utilization; and (b) increased capacity utilization may also necessitate the high-level digital development of enterprises, weakening unnecessary trade resistance and assisting them in improving production efficiency. The instrumental variables approach was used in this study to mitigate the reverse causality of the aforementioned potential endogenous effects on our findings.

Following Huang et al. [33], the instrumental variables for firm digitalization in this study were derived from 1984 postal data for each city. This approach was adopted primarily because the communication methods used in previous development processes at a firm's location tend to influence that firm's application and acceptance of information technology in terms of technology level and social preferences during the sample period, which satisfies the correlation condition. Postal and telecommunications services, as forms of social infrastructure, primarily facilitate communication for the population and do not directly influence firm capacity utilization processes, so they satisfy the exogeneity condition. Furthermore, because the 1984 postal and telecommunications information for each city is cross-sectional data that are difficult to use directly as instrumental variables for panel data, we followed Yuan Chun et al. [20] and used the cross-product term of the number of individuals with internet access in the nation with a lag of one period and the number of fixed telephones per 10,000 people at the end of 1984 as an instrumental variable for the digitalization progress of enterprises in the current period. The results in the first column of Table 4 reveal that the explanatory variable coefficients are significantly positive at the 5% level, thereby demonstrating the validity of this study's primary findings.

## Alternative measures of digital transformation

Several methods were employed in this study to arrive at alternative measures of a company's digital transformation. The quantification of digital transformation in the sample firms is based on the frequency of special words of a digital nature appearing in firm annual statements, which reflects the importance of digital transformation. However, this quantification does not consider firm investment in the development of digital infrastructure. Therefore, by using digital technology applications and digital technology devices as proxy variables for enterprise digital transformation in our robustness tests, we can examine the conditions under which digital transformation can be achieved. Specifically, these variables represent the number of people employed in the information transmission, computer services, and software industries (10 million) and the number of broadband access subscribers (100 million) each year in the cities containing listed companies. According to the results in the second and third columns of Table 4, the explanatory variable coefficients are significantly positive, indicating that the results of this study are robust. In particular, the degree of digital transformation, which has been replaced with the number of broadband subscribers, has a significant coefficient at the 1% level, indicating that the direction and significance of the regression coefficients remain the same as previously stated. This supports the robustness of our primary findings.

## Robustness tests

OC, which is the inverse of capacity utilization, was used to replace the explanatory variable as an alternative measure. The fourth column in Table 4 reveals that the explanatory variable coefficient is significantly negative, and because the alternative measure is the inverse of

**Table 4. Results of robustness tests.**

| Variables | (1) | (2) | (3) | (4) | (5) |
|---|---|---|---|---|---|
| | CU | CU | CU | OC | CU |
| DT_w1 | 4.567** | | | -0.099* | 0.469** |
| | (2.106) | | | (0.055) | (0.198) |
| news | | 0.009* | | | |
| | | (0.005) | | | |
| Int us | | | 10.526*** | | |
| | | | (3.068) | | |
| size2 | −0.596*** | −0.997*** | −0.992*** | 0.865*** | −1.043*** |
| | (0.038) | (0.071) | (0.071) | (0.024) | (0.071) |
| roa | 3.818*** | 1.668*** | 1.66*** | −1.183*** | 1.785*** |
| | (0.603) | (0.444) | (0.444) | (0.109) | (0.444) |
| lev | 4.818*** | 2.612*** | 2.641*** | −0.15 | 2.707*** |
| | (0.275) | (0.33) | (0.329) | (0.095) | (0.329) |
| age | 0.258*** | −0.341*** | −0.342*** | 0.113*** | −0.334*** |
| | (0.067) | (0.118) | (0.118) | (0.032) | (0.117) |
| soe | 0.195 | −0.114 | −0.106 | −0.093 | −0.277 |
| | (0.135) | (0.265) | (0.265) | (0.075) | (0.255) |
| Constants | 2.614*** | 9.015*** | 8.743*** | 13.455*** | 9.363*** |
| | (0.47) | (0.568) | (0.575) | (0.176) | (0.565) |
| Observations | 15087 | 14751 | 14751 | 8161 | 15909 |
| Adj-R$^2$ | 0.061 | 0.69 | 0.691 | 0.802 | 0.686 |

**Note**: t-statistics adjusted for firm-level clustering are in parentheses.

capacity utilization, this indicates that digitalization significantly suppresses corporate overcapacity and supports the robustness of our primary findings. Region- and industry-fixed effects are included in the fifth column, which are the interactive fixed effects of the province and year, and industry and year of the listed companies. The explanatory variable coefficients are significant at the 5% level, which supports the robustness of our primary findings.

## Additional analyses

### Mechanism testing and analysis

The analyses presented thus far reveal that the mechanism of digital transformation for promoting enterprise capacity utilization is the promotion of enterprise innovation and production specialization, which increases the efficiency of enterprise resource utilization. Therefore, VSI and technological innovation were included in our analysis of H1. The corresponding regression results are presented in Table 5.

The results in the first column of Table 5 reveal that enterprise digital transformations are significantly and positively related to production specialization at the 1% level, which can be interpreted as moderate digital transformation assisting enterprises in exploiting their comparative advantages in resources and technology, improving their transaction environment, reducing their internal control costs, and promoting their specialization. The second column reveals that the VSI coefficient is significantly positive at the 1% level, whereas the DT coefficient decreases in both value and significance. Therefore, a combined-effects test using the stepwise regression method confirms that digital transformation improves capacity utilization by encouraging firms to achieve production specialization. The third column reveals that at the 1% level, firm digital

**Table 5. Heterogeneity analysis results.**

| | (1) | (2) | (3) | (4) | (5) |
|---|---|---|---|---|---|
| | VSI | CU | Inn | CU | CU |
| DT | 0.023*** | 0.399** | 0.209*** | 0.102 | 0.088 |
| | (0.007) | (0.18) | (0.05) | (0.064) | (0.064) |
| VSI | | 1.633*** | | | 0.71*** |
| | | (0.221) | | | (0.087) |
| Inn | | | | 0.07*** | 0.069*** |
| | | | | (0.014) | (0.014) |
| size | 0.002 | −1.056*** | 0.321*** | −0.857*** | −0.851*** |
| | (0.003) | (0.065) | (0.021) | (0.028) | (0.028) |
| roa | 0.145*** | 1.698*** | 0.434** | 4.53*** | 4.316*** |
| | (0.017) | (0.432) | (0.212) | (0.273) | (0.273) |
| lev | 0.122*** | 2.494*** | 0.223** | 1.735*** | 1.612*** |
| | (0.012) | (0.311) | (0.096) | (0.123) | (0.124) |
| age | 0.002 | −0.343*** | 0.021 | −0.008 | −0.015 |
| | (0.004) | (0.106) | (0.031) | (0.04) | (0.04) |
| soe | −0.018** | −0.111 | −0.077 | −0.121 | −0.1 |
| | (0.009) | (0.24) | (0.067) | (0.086) | (0.086) |
| Constants | 0.473*** | 8.633*** | −.406** | 7.115*** | 6.752*** |
| | (0.02) | (0.526) | (0.166) | (0.214) | (0.217) |
| Observations | 15912 | 15912 | 10589 | 10589 | 10589 |
| Adj-R$^2$ | 0.708 | 0.674 | 0.806 | 0.855 | 0.856 |

**Note**: t-statistics adjusted are in parentheses.

transformation significantly promotes technological innovation, likely because digital transformation stimulates product and process innovation, which promotes firm technological innovation. The fourth column reveals that the coefficient of firm technological innovation is significantly positive at the 1% level, but the coefficient of digital transformation decreases in value and is not significant, indicating that the effect of DT on firm capacity utilization is translated into technological innovation, which in turn means that there is technological innovation promoting the capacity utilization of firms. Such technological innovation has a well-mediated impact. The fifth column incorporates all mediating variables into the model, and one can see that the VSI coefficient is significantly greater than the coefficient of technological innovation for the two mediating variables that are also significant at the 1% level. This indicates that both increased specialized production capacity and technological innovation can positively influence firm capacity utilization, but the effect of specialized production capacity is stronger. However, when the explanatory variable coefficients (DT) are compared, the effect of digital transformation is insignificant, despite having a positive impact on capacity utilization.

## Heterogeneity tests

As discussed in the previous theoretical analysis, digitalization alters the internal and external environments of enterprises, thereby affecting their capacity utilization level. Specifically, digitalization improves enterprise capacity utilization by promoting innovation and production specialization. However, the levels of enterprise capacity utilization and digital transformation vary across enterprises and industries, which may result in heterogeneity in the effects of digital transformation on enterprise capacity utilization. Therefore, we further subdivided our data sample to examine heterogeneity based on corporate governance, company size, and industry dimensions.

### Heterogeneity analysis based on enterprise size

The first and second columns in Table 6 present the estimation results for two subsamples: firms with less-than-average capital size and firms with greater-than-average capital size. The

**Table 6. Regression results of mechanism analysis.**

|  | (1) | (2) | (3) | (4) | (5) | (6) |
|---|---|---|---|---|---|---|
|  | CU | CU | CU | CU | CU | CU |
| DT | −0.364 | 0.219*** | 0.187 | 0.484** | 0.151** | 0.175 |
|  | (0.312) | (0.047) | (0.308) | (0.217) | (0.075) | (0.438) |
| size | −2.238*** | −.473*** | −1.312*** | −.914*** | −.711*** | −1.913*** |
|  | (0.104) | (0.022) | (0.101) | (0.091) | (0.025) | (0.169) |
| roa | 2.225** | 0.539*** | 3.903*** | 1.194*** | 1.355*** | 3.843*** |
|  | (1.074) | (.09) | (.965) | (.455) | (.137) | (1.489) |
| lev | 1.949*** | .835*** | 2.58*** | 2.324*** | 1.223*** | 5.608*** |
|  | (0.529) | (0.087) | (0.491) | (0.407) | (0.108) | (0.896) |
| age | −0.229 | 0.062** | −0.227 | −0.506*** | −0.012 | −0.257 |
|  | (0.217) | (0.029) | (0.208) | (0.138) | (0.036) | (0.313) |
| soe | −0.063 | −0.162** | −0.095 | −0.143 | −0.092 | −0.105 |
|  | (0.323) | (0.079) | (0.292) | (0.486) | (0.079) | (0.694) |
| Constants | 21.146*** | 3.943*** | 11.985*** | 7.992*** | 6.192*** | 15.614*** |
|  | (1.024) | (0.146) | (0.918) | (0.667) | (0.193) | (1.344) |
| Observations | 7844 | 7830 | 8125 | 7411 | 8629 | 4234 |
| Adj-R$^2$ | 0.844 | 0.856 | 0.765 | 0.591 | 0.712 | 0.819 |

**Note**: t-statistics adjusted are in parentheses.

estimated coefficient of DT in the sample of firms with smaller capital size is significant at the 1% level. However, the regression results for the sample of firms with larger capital are not significant and the coefficients are negative. This suggests that the digital transformation of firms with smaller capital can effectively contribute to an increase in their capacity utilization by stimulating firm innovation and enhancing production specialization. However, in firms with larger capital, digital transformation has no significant impact on capacity utilization. There may be greater resistance to alleviating overcapacity in such enterprises, which reduces the impact of digital transformation. Therefore, larger capital-intensive firms find it challenging to improve their capacity utilization through digital transformation.

### Heterogeneity analysis based on corporate governance level.

The third and fourth columns in Table 6 present the subsample regression results for firms with two different levels of governance. The estimated DT coefficient in the sample with poorer governance is significantly positive at the 1% level. Although numerically positive, the regression results for the sample with higher levels of governance are not significant. This suggests that firms with lower levels of governance will benefit from digital transformation through the stimulation of innovation and improved specialized production, which will lead to an increase in capacity utilization. Digital transformation can compensate for deficits in capacity utilization caused by a lack of corporate governance in poorly governed firms by improving their resource allocation capabilities. Conversely, the impact of digital transformation on capacity utilization is not significant in firms with a sound governance system. This may be because companies with higher governance levels have greater resource allocation capabilities, making the effects of further improvement in capacity utilization levels induced by digital transformation less pronounced. Therefore, corporate governance can be considered as a substitute for digital transformation.

### Heterogeneity analysis based on industry.

The fifth and sixth columns in Table 6 present the regression results for manufacturing and service firms, respectively. The estimated DT coefficient for the sample of manufacturing firms is significantly positive at the 5% level. However, the regression results for the sample of service firms are insignificant, despite being positive. This suggests that, for manufacturing firms, digital transformation can contribute to increased capacity utilization through corporate innovation and improved production specialization. In contrast, the role of digital transformation in improving the overcapacity of service firms is insignificant. This is most likely because manufacturing companies have significant overcapacities and potential to improve capacity utilization. In contrast, service firms have lower transaction information friction and higher capacity utilization, making the impact of digital transformation less pronounced.

## Discussion

Overcapacity has resulted in significant resource misallocation, and waste remains a major issue for economic growth that must be addressed. Most previous studies on paths and methods to enhance capacity utilization have been conducted from a macro perspective based on government control, business environments, state-owned enterprise soft budgets, and market-oriented mechanisms and have seldom investigated the role of digital transformation at the enterprise level. The deep integration of real economies and digitalization has become a major trend and an inevitable result of future development since the advent of the digital era. The healthy development of the digital economy promotes the construction of a modern economic system. As a new type of production factor, data have a significant impact on changes in

traditional production methods. The digital economy is not only a new economic growth point, but also a fulcrum for transforming and upgrading traditional industries, and it can be an important engine for constructing a modernized economic system with high innovation, strong penetration, and widespread coverage. In this regard, no systematic studies have been conducted on how the digital transformation of enterprises improves overcapacity and few studies have extensively examined the alternative roles of digital transformation and corporate governance.

Unlike previous studies that investigated capacity utilization governance from macro and regional perspectives, we examined the impact of digital transformation on capacity utilization from an enterprise-level micro perspective. We elucidated the theoretical mechanisms by which the characteristics of digital transformation affect enterprise innovation and resource allocation efficiency. In other words, we examined how enterprise digital transformation can effectively promote enterprise technological innovation, improve resource allocation efficiency, strengthen a firm's production specialization, and effectively alleviate overcapacity. Furthermore, this study's findings demonstrate that the effect of digital transformation on capacity utilization has an inverted U shape and that relying solely on digital transformation to suppress enterprise overcapacity is ineffective. The impact of digital transformation on capacity utilization was found to be more significant for enterprises with smaller capital and poorer governance. Furthermore, based on the lower likelihood of information mismatch in service firms, digital transformation appears to alleviate overcapacity more effectively in manufacturing enterprises than in service organizations.

It should be pointed out that although this study adopted a text analysis method of machine learning to study the overall scenario of enterprise digital transformation, we have not yet derived an accurate description of the digital aspects of enterprises (e.g., innovation, production, sales). More work is necessary to characterize the detailed aspects of digital transformation and reveal the impact mechanisms of digital transformation on the external transaction costs and internal control costs of enterprises.

## Supporting information

**S1 Data. Collecting data.**
(ZIP)

## Acknowledgments

We wish to thank Shi Junwei, Liu Xinglin, Hu Shan, and Guo Yanbing. We also wish to thank the Center for Industrial Economics of the Zhongnan University of Economics and Law.

## Author Contributions

**Conceptualization:** Ning Zhao.

**Data curation:** Ning Zhao.

**Formal analysis:** Ning Zhao.

**Funding acquisition:** Jianxin Ren.

**Investigation:** Jianxin Ren.

**Methodology:** Jianxin Ren.

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
