## [Decision Letter · Decision Letter 0]

17 Oct 2022

PONE-D-22-19462Enterprise Digital Transformation and Capacity UtilizationPLOS ONE

Dear Dr. Ning Zhao,

Thank you for submitting your manuscript to PLOS ONE. After careful consideration, we feel that it has merit but does not fully meet PLOS ONE’s publication criteria as it currently stands. Therefore, we invite you to submit a revised version of the manuscript that addresses the points raised during the review process.

We look forward to receiving your revised manuscript.

Kind regards,

Bo Huang

Academic Editor

PLOS ONE

Journal Requirements:

Reviewers' comments:

Reviewer's Responses to Questions

**Comments to the Author**

1. Is the manuscript technically sound, and do the data support the conclusions?

Reviewer #1: Yes

Reviewer #2: No

2. Has the statistical analysis been performed appropriately and rigorously? 

Reviewer #1: Yes

Reviewer #2: No

3. Have the authors made all data underlying the findings in their manuscript fully available?

Reviewer #1: No

Reviewer #2: Yes

4. Is the manuscript presented in an intelligible fashion and written in standard English?

Reviewer #1: No

Reviewer #2: No

5. Review Comments to the Author

Reviewer #1: The paper needs some revisions before publication.

In the paper, the authors used the data of Chinese listed companies and showed that enterprises' digital transformation has a great significance in increasing the rate of enterprises' capacity utilization.

1- There are a huge number of published studies available in the field. How this study significantly contributes to the literature?

2- Please cite papers from the different areas, as the author mainly focuses on Chinese literature such as “China has made significant achievements in economic growth”. While many countries, such as India, the USA, and Israel, are adopting digital transformation.

3- Expand the introduction by supporting the difference between digital and conventional transformation and cite some high-quality papers to support the idea.

4- Scientific literature is lacking, please include more references related to the study.

5- There are many characteristics of the overcapacity industry, but I can see only three have been mentioned in

section II.

6- Section II, Frist Paragraph, what do the distinctive Chinese characteristic please specify, and explain how these characteristics result in overcapacity.

7- Section II first paragraph, “The 2022 Government work report” cite the reference.

8- Please include the references to support your result and conclusion. Interpretation needs to be simplified and discussed with references.

9- Include the limitation of the study.

10- There are many typing and grammatical errors, which needs to be corrected.

Reviewer #2: 1 Summary

This paper studies the impact of digitalization on the capacity utilization of firms through the path

of collaborative relationships and efficient innovations. Through literature reviews and qualitative

analysis, the author abstracts three core hypotheses and empirical studies using the data from

A-share listed companies in China find that the impact mechanism of firms’ digital transformation

on its capacity utilization is nonlinear U-shaped. Endogenous analysis, robust analysis and

heterogeneous analysis further confirm this conclusion.

2 Main comments

1. The English writing of this article is not up to standard, there are many grammatical errors, and

the expression is not authentic, which is far from meeting the requirements of publication. There

are unnecessary long difficult sentences in the article, and too many “and” are inappropriately

used. I suggest the author(s) carefully revise the paper and submit it to a Chinese journal;

otherwise the language should be greatly improved.

2. Many of the discussions and conclusions in this paper are neither supported by theoretical

literature nor supported by theoretical model analysis, which greatly affects the credibility of this

paper, e.g. the paper concludes that “digital transformation of enterprise will promote innovation,

improve resource allocation and production capacity, so as to effectively curb overcapacity”. How

to curb overcapacity through digital transformation is an interesting question, but there is no

model or theoretical analysis in this paper, so the mechanism is not well explored. As a result,

many conclusions are just like basic judgments. I suggest the authors write a model to explain how

digital transformation can better curb overcapacity compared with other ways.

3. Three hypotheses are not well organized. As far as I’m concerned, H1, H2 and H3 are about a

same thing: how digitalization promotes capacity utilization, so the main topic can be summarized

into one hypothesis, which can be treated as the paper’s main proposition and contribution. As

stated in point 2, the authors can write a theoretical model to see whether the U-shaped nonlinear

relationship can be derived from the model.

3 Minor comments and typos

1. The authors cite so many Chinese journals, and there are no more than 8 articles cited from

international authoritative journals.

2. The citing format in the article is wrong.

3. The formula is not fully displayed, although I can guess what it is.

6. PLOS authors have the option to publish the peer review history of their article (what does this mean?). If published, this will include your full peer review and any attached files.

Reviewer #1: No

Reviewer #2: No

---

## [Author Response · Author response to Decision Letter 0]

7 Dec 2022

Dear reviewers,

 Thank you so much for your comments and professional advice. These comments helped to improve the academics of my article. Following your suggestions and requests, we have made corrections to the revised manuscript. At the same time, the grammar of the manuscript has been reviewed and edited. I hope our work can be improved again. Moreover, we would like to show the details as follows:

Reviewer 1＃

1- There are a huge number of published studies available in the field. How this study significantly contributes to the literature?

 The author’s answer: The reviewer makes a valid point that there are numerous articles on digital transformation and a relatively large number of articles studying capacity, but almost no articles examining the effect of digital transformation on capacity utilization. Therefore, this study opens up a contemporary research perspective on digital transformation. Moreover, most of the studies on digital transformation have been conducted from a positive perspective to investigate its impact mechanisms. Based on this, the research presented in this paper adds a study on the negative effects of digital transformation to fill the gap in research thinking. In addition, current measurements and characterization of capacity utilization are usually based on a macro level, e.g., country, region, etc., whereas the calculations of capacity utilization in this paper are at a more microscopic level, which leads to more accurate conclusions. (See I. Introduction in the revised manuscript for details)

2-Please cite papers from the different areas, as the author mainly focuses on Chinese literature such as “China has made significant achievements in economic growth”. While many countries, such as India, the USA, and Israel, are adopting digital transformation.

 The author’s answer: The reviewer’s comments are professional. With the development of science and technology, digital transformation is advocated in many countries, and there are many geographically distributed companies actively implementing digital management and services. However, there are some fundamental differences between these countries and regions and Chinese companies, such as over-building, over-investment, which hindered the efficient use of their resources and even prevents healthy economic development, and low capacity utilization and over-capacity. On top of this, the Chinese government urgently needs to find a tool to reverse this passive situation, so the digital transformation of local companies is particularly important. For these reasons, it is a more appropriate choice to study the impact of digital transformation on capacity ratios by analyzing the mechanism of its effect using data from Chinese-listed companies.

3-Expand the introduction by supporting the difference between digital and conventional transformation and cite some high-quality papers to support the idea.

 The author’s answer: Your advice is perfectly professional. Based on your comments, I have added a high-level article related to the topic in the Introduction section as appropriate, as well as a more in-depth analysis and categorization of the current research on digital transformation. I have identified the current research gap in the analysis of the role of digitalization mechanisms and explored them as a research focus.

 4- Scientific literature is lacking, please include more references related to the study.

 The author’s answer: Thank you for your corrections. It has been added.

 5- There are many characteristics of the overcapacity industry, but I can see only three have been mentioned in section II.

 The author’s answer: Thank you for the correction. Excess capacity may exist at different stages of industry development. However, the causes of overcapacity are very different. For example, overcapacity is caused by competition in the early stages of industry development, or overcapacity is caused by declining market demand in sunset industries. Since this is not the focus of this paper, I have not highlighted it in the paper after further consideration.

 6- Section II, First Paragraph, what do the distinctive Chinese characteristic please specify, and explain how these characteristics result in overcapacity.

 The author’s answer: Thank you for the reminder, and based on your suggestion, I have added an introduction to Chinese firm features in the first section. The term capacity utilization, which serves as the focal explanatory variable in this study, has not been prominent in other country studies. This may be related to the historical process of overcapacity formation. It is well known that the most direct cause affecting capacity utilization is over-investment, and there are many causes that induce over-investment. In compiling the literature, I found that the main factors contributing to overcapacity in academia are broadly divided into three categories: 1. Industry competition, excessive competition in the industry, competitive strategies adopted by enterprises to deter and threaten their competitors, which triggers the expansion of production capacity and induces overcapacity; 2. Development dynamics, the frequent occurrence of investment surges in developing countries, and the role of market interest rates and price adjustment mechanisms are emphasized in the theory. The theory is that in developing countries, a large influx of capital into certain industries, i.e., an "investment boom", triggers the phenomenon of increasing overcapacity; 3. Government-business relationship, the most commonly studied theory in China, argues that local officials want to provide land and financing investments to local enterprises in the promotion race, resulting in the over-expansion of capacity and causing under-utilization of capacity.

 7- Section II first paragraph, “The 2022 Government work report” cite the reference.

 The author’s answer: Sorry for this error due to my negligence. Thank you for pointing it out, it has been corrected.

 8- Please include the references to support your result and conclusion. Interpretation needs to be simplified and discussed with references.

 The author’s answer: Thank you so much for your suggestion. I have made changes.

 9- Include the limitation of the study.

 The author’s answer: In light of your comments and suggestions, we have added the limitations of the study in the "1 Introduction" section.

 10- There are many typing and grammatical errors, which needs to be corrected.

The author’s answer: Thank you for your advice. The grammatical problems have been corrected one by one.

Reviewer 2＃

 Main comments

1.The English writing of this article is not up to standard, there are many grammatical errors, and the expression is not authentic, which is far from meeting the requirements of publication. There are unnecessary long difficult sentences in the article, and too many “and” are inappropriately used. I suggest the author(s) carefully revise the paper and submit it to a Chinese journal; otherwise the language should be greatly improved.

 The author’s answer: Thank you for your criticism. I re-read the article carefully and revised it, and have made sufficient repairs to eliminate grammatical errors, correct the use of "and", and make changes to the long and difficult sentences.

2.Many of the discussions and conclusions in this paper are neither supported by theoretical literature nor supported by theoretical model analysis, which greatly affects the credibility of this paper, e.g. the paper concludes that “digital transformation of enterprise will promote innovation, improve resource allocation and production capacity, so as to effectively curb overcapacity”. How to curb overcapacity through digital transformation is an interesting question, but there is no model or theoretical analysis in this paper, so the mechanism is not well explored. As a result,many conclusions are just like basic judgments. I suggest the authors write a model to explain how digital transformation can better curb overcapacity compared with other ways.

 The author’s answer: The reviewer makes very good points and your suggestions are very professional. The logic and theory of the article have been further refined and reconstructed with your suggestions. In the introduction section, I appropriately introduce the many high-quality works in the literature to sort out the objective factors the influence capacity utilization at a more specific level, as well as to point out the reasons for selecting Chinese-listed companies as the target of the study. In addition, in sorting out the impact and mechanism of action of current enterprise capacity utilization, I discovered its research gap - on capacity utilization, which provides ideas for further analysis and paves the way for the theoretical analysis and research hypothesis in the later sections.

In the theoretical analysis sections, I adopt the previous analysis framework and introduce the main assumptions of this paper based on reading a large amount of high-level literature. The theoretical analysis is taken from the classical literature, which enhances the credibility of the article. In the analysis of the impact of the digital transformation of firms on capacity utilization, two mediating variables, firm innovation, and specialized production, are introduced in this paper. Among them, for the hypothesis that digital transformation drives enterprise innovation, I explore in more depth the specific paths and implementation capabilities of innovation affecting enterprise capacity by specifically analyzing the factors that affect the role of capacity utilization and enumerating the mechanisms of impact of digital transformation. I think this can clearly show the mechanism of enterprise innovation as a mechanism. In the analysis of the specialization and division of labor and the innovative and important intermediary of specialized production of enterprises, I have explored it from the perspective of enterprise expansion and development. I am aware of the importance of clarifying the law of firm development for the study of this paper. Therefore, by reviewing the literature, I found that the horizontal expansion of enterprises inevitably promotes the efficient use of resources. Moreover, digital transformation can expand the production boundaries of enterprises and enhance specialized production capacity by reducing production costs and management costs. These factors can improve the efficiency of resource utilization, which in turn has a positive effect and impact on capacity utilization. Therefore, the specialized production segmentation of enterprises is presented in this section as one of the highlights of this study.

With these in-depth explorations and studies, the first hypothesis – the positive impact of digital transformation on enterprises – is presented here in this study. The theoretical framework is reconstructed to make the logic of the paper seem more reasonable and to make the theory of the paper more accessible and acceptable. (See I. Introduction and II. Theoretical analysis in the revised version for more details)

3.Three hypotheses are not well organized. As far as I’m concerned, H1, H2 and H3 are about a same thing: how digitalization promotes capacity utilization, so the main topic can be summarized into one hypothesis, which can be treated as the paper’s main proposition and contribution. As stated in point 2, the authors can write a theoretical model to see whether the U-shaped nonlinear relationship can be derived from the model.

 The author’s answer: We would like to thank the reviewers for their very professional and pertinent suggestions. Following the suggestions of the reviewers, after further deliberation and careful consideration, I have made the following changes to the hypothesis section of the paper – Hypothesis 1 and Hypothesis 3 are combined, as Hypothesis 2 is retained as a research innovation of this paper.

The specific reasons are as follows: While these assumptions are all about the impact of digital transformation on capacity utilization, there are positive and negative mechanisms of action. A careful study of the literature reveals that the mechanism of action of digital transformation on capacity utilization has not only mediated the positive effects of innovation and specialized production, but also some negative adverse effects may exist. There is no guarantee that digital transformation will necessarily lead to efficient resource utilization. In the case of the communications industry, for example, the overuse of digital services has a more significant negative impact on data-intensive manufacturing. The lack of a better institutional design for advancing digital transformation in the initial stages of digital transformation will lead to a failure to truly improve the efficiency of resource utilization in the enterprise. However, contemporary research has paid little attention to the negative impact of digital transformation on business development, that is, almost all research has focused on the positive impact of digital transformation on business. Therefore, the U-shaped character of the impact of digital transformation on the capacity utilization of firms can be used as an innovative analysis for the paper's research.

I strongly endorse the professional suggestions of the reviewers. Indeed, at the beginning of this research, I also considered whether to construct a theoretical model to explain this mechanism. By referring to a large body of literature and comparing analyses, I found that constructing a theoretical model does not explain the role of digital transformation as well as a theoretical analysis does, nor does it visually demonstrate the impact of digitalization in the enterprise. In general, the mechanism of digital transformation can be more generally understood and described by theoretical analysis, and therefore, after repeated consideration, I have chosen theoretical analysis to describe this mechanism of influence.

Minor comments and typos

1. The authors cite so many Chinese journals, and there are no more than 8 articles cited from international authoritative journals.

The author’s answer: This has been increased to 28 articles.

2. The citing format in the article is wrong.

The author’s answer: Thank you for the kind corrections. They have been corrected.

3. The formula is not fully displayed, although I can guess what it is.

 The author’s answer: I am very sorry for this error, which was due to my oversight. I thank the reviewers for their corrections and have made adjustments to confirm that it is fully displayed.

Thank you for giving us the opportunity to submit a revised draft of this manuscript for publication in Plos One. We appreciate the time and effort that you and the reviewers have devoted to providing feedback on our manuscript and are grateful for insightful comments and valuable improvements to our manuscript. We have incorporated most of the suggestions made by the reviewers. These changes are highlighted in the manuscript. 

According to the reviewer’s comments, we have revised the manuscript extensively. If there are any other modifications we could make, we would like very much to modify them and we appreciate your help. We hope that our manuscript will be considered for publication in your journal. Thank you so much for your help. 

Yours sincerely,

Ning Zhao

November 20, 2022

Zhongnan University of Economics and Law

---

## [Decision Letter · Decision Letter 1]

15 Feb 2023

PONE-D-22-19462R1Impact of enterprise digital transformation on capacity utilization: Evidence from ChinaPLOS ONE

Dear Dr. Ning Zhao,

Thank you for submitting your manuscript to PLOS ONE. After careful consideration, we feel that it has merit but does not fully meet PLOS ONE’s publication criteria as it currently stands. Therefore, we invite you to submit a revised version of the manuscript that addresses the points raised during the review process.

We look forward to receiving your revised manuscript.

Kind regards,

Bo Huang

Academic Editor

PLOS ONE

Journal Requirements:

Reviewers' comments:

Reviewer's Responses to Questions

**Comments to the Author**

1. If the authors have adequately addressed your comments raised in a previous round of review and you feel that this manuscript is now acceptable for publication, you may indicate that here to bypass the “Comments to the Author” section, enter your conflict of interest statement in the “Confidential to Editor” section, and submit your "Accept" recommendation.

Reviewer #1: All comments have been addressed

Reviewer #2: (No Response)

2. Is the manuscript technically sound, and do the data support the conclusions?

Reviewer #1: Yes

Reviewer #2: Partly

3. Has the statistical analysis been performed appropriately and rigorously? 

Reviewer #1: Yes

Reviewer #2: Yes

4. Have the authors made all data underlying the findings in their manuscript fully available?

Reviewer #1: No

Reviewer #2: No

5. Is the manuscript presented in an intelligible fashion and written in standard English?

Reviewer #1: Yes

Reviewer #2: No

6. Review Comments to the Author

Reviewer #1: Please include the detailed limitation of studies, I can see only one point in last paragraph. Please describe and include all the limitation.

Thanks

Reviewer #2: The author has revised and improved the draft according to the comments of the first review, but there are still several following aspects that need further revisions:

1. There are still grammatical errors in the abstract, introduction, theory and hypotheses, discussion part of the revised version, which limit the clearness, coherence and readability of this paper. I revised the manuscript directly in the text sentence by sentence. Please refer to the yellow mark on pages 46-82 of the attached PDF file;

2. The formulas are not successfully displayed, so I can't judge whether the formulas are correct and reasonable;

3. the note "t-statistics adjusted for firm-level clustering are in parentheses; ***, **, and * indicate significance at

the 1%, 5%, and 10% levels respectively" should be shown in Table 3, 4, 5, and 6;

4. Please reconfirm that the citation format and the reference format meet the requirements of the journal.

7. PLOS authors have the option to publish the peer review history of their article (what does this mean?). If published, this will include your full peer review and any attached files.

Reviewer #1: No

Reviewer #2: No

---

## [Author Response · Author response to Decision Letter 1]

28 Feb 2023

Dear reviewers:

 Thank you for your letter and for the reviewers' comments concerning our manuscript. Those comments are all valuable and very helpful for revising and improving our paper,as well as the important guiding significance to our researches. We have studied comments carefully and have made correction which we hope meet with approval. Revised portion are marked in the paper. The main corrections in the paper and the responds to the reviewer's comments are as flowing: Responds to the reviewer's comments:

Reviewer #1: Please include the detailed limitation of studies, I can see only one point in last paragraph. Please describe and include all the limitation.

Thanks

 The author’s answer: Thank you very much for your professional advice, which has helped me a lot in my research. According to your suggestion, I have supplemented the description of research limitations (see I. Introduction for details). Thank you again for your guidance on this article, which has inspired my research and writing.

Reviewer #2: The author has revised and improved the draft according to the comments of the first review, but there are still several following aspects that need further revisions:

1. There are still grammatical errors in the abstract, introduction, theory and hypotheses, discussion part of the revised version, which limit the clearness, coherence and readability of this paper. I revised the manuscript directly in the text sentence by sentence. Please refer to the yellow mark on pages 46-82 of the attached PDF file;

 The author’s answer: Thank you very much for your professional and meticulous attitude. Your careful and responsible revision has motivated me to do research. Under your revision, I have re-read and modified the grammar of the full text in order to reach the standard. The article retains your professional and exquisite guidance. Thank you for your continued contributions to this manuscript, and I will work with absolute seriousness to improve the content of this paper.

2. The formulas are not successfully displayed, so I can't judge whether the formulas are correct and reasonable;

 The author’s answer: I am very sorry for the situation. When I submitted the manuscript, I specifically checked the formulas and made sure they could be shown. Is it a matter of version? The formulas are similar to the content of the first edition. No essential changes were made. Please rest assured that I will contact the editorial department to ensure that the formulas are properly presented.

3. the note "t-statistics adjusted for firm-level clustering are in parentheses; ***, **, and * indicate significance at the 1%, 5%, and 10% levels respectively" should be shown in Table 3, 4, 5, and 6;

 The author’s answer: Thank you very much for your reminding, I have made the changes and made sure they are fully displayed.

4. Please reconfirm that the citation format and the reference format meet the requirements of the journal.

 The author’s answer: Thank you for your kind correction. This section has been corrected as required by the format.

Thank you for giving us the opportunity to submit a revised draft of this manuscript for publication in Plos One. We appreciate the time and effort that you and the reviewers have devoted to providing feedback on our manuscript and are grateful for insightful comments and valuable improvements to our manuscript. We have incorporated most of the suggestions made by the reviewers. These changes are highlighted in the manuscript. 

According to the reviewer’s comments, we have revised the manuscript extensively. If there are any other modifications we could make, we would like very much to modify them and we appreciate your help. We hope that our manuscript will be considered for publication in your journal. Thank you so much for your help. 

Yours sincerely,

Ning Zhao

February 28, 2023

Zhongnan University of Economics and Law

---

## [Editor Report · Decision Letter 2]

6 Mar 2023

Impact of enterprise digital transformation on capacity utilization: Evidence from China

PONE-D-22-19462R2

Dear Dr. Ning Zhao,

We’re pleased to inform you that your manuscript has been judged scientifically suitable for publication and will be formally accepted for publication once it meets all outstanding technical requirements.

Kind regards,

Bo Huang

Academic Editor

PLOS ONE

---

## [Editor Report · Acceptance letter]

8 Mar 2023

PONE-D-22-19462R2 

Impact of enterprise digital transformation on capacity utilization: Evidence from China 

Dear Dr. Zhao:

I'm pleased to inform you that your manuscript has been deemed suitable for publication in PLOS ONE. Congratulations! Your manuscript is now with our production department. 

Kind regards, 

on behalf of

Professor Bo Huang 

Academic Editor

PLOS ONE